



# Estimating global cropland production from 1961 to 2010

Pengfei Han[1*], Ning Zeng[1,2*], Fang Zhao[2,3], Xiaohui Lin[4]

[1]State Key Laboratory of Numerical Modeling for Atmospheric Sciences and Geophysical Fluid Dynamics, Institute of Atmospheric Physics, Chinese Academy of Sciences, Beijing 100029, China

[2]Department of Atmospheric and Oceanic Science, and Earth System Science Interdisciplinary Center, University of Maryland, College Park, Maryland 20742, USA

[3]Potsdam Institute for Climate Impact Research, Potsdam, Brandenburg 14473, Germany

[4]State Key Laboratory of Atmospheric Boundary Layer Physics and Atmospheric Chemistry, Institute of Atmospheric Physics, Chinese Academy of Sciences, Beijing 100029, China

*Correspondence to*:

Ning Zeng (zeng@lasg.iap.ac.cn);

Pengfei Han (pfhan@mail.iap.ac.cn)



**Abstract.** Global cropland net primary production (NPP) has tripled over the last fifty
years, contributing 17-45 % to the increase of global atmospheric $CO_2$ seasonal
amplitude. Although many regional-scale comparisons have been made between
statistical data and modelling results, long-term national comparisons across global
croplands are scarce due to the lack of detailed spatial-temporal management data.
Here, we conducted a simulation study of global cropland NPP from 1961 to 2010
using a process-based model called VEGAS and compared the results with Food and
Agriculture Organization of the United Nations (FAO) statistical data on both
continental and country scales. According to the FAO data, the global cropland NPP
was 1.3, 1.8, 2.2, 2.6, 3.0 and 3.6 PgC yr$^{-1}$ in the 1960s, 1970s, 1980s, 1990s, 2000s
and 2010s, respectively. The VEGAS model captured these major trends at global and
continental scales. The NPP increased most notably in the U.S. Midwest, Western
Europe and the North China Plain, and increased modestly in Africa and Oceania.
However, significant biases remained in some regions such as Africa and Oceania,
especially in temporal evolution. This finding is not surprising as VEGAS is the first
global carbon cycle model with full parameterization representing the Green
Revolution. To improve model performance for different major regions, we modified
the default values of management intensity associated with the agricultural Green
Revolution differences across various regions to better match the FAO statistical data
at the continental level and for selected countries. Across all the selected countries,
the updated results reduced the root mean square error (RMSE) from 19.0 to 10.5 TgC
yr$^{-1}$ (~45 % decrease). The results suggest that these regional differences in model
parameterization are due to differences in social-economic development. To better
explain the past changes and predict the future trends, it is important to calibrate key
parameters at regional scales and develop datasets for land management history.



## 1 Introduction

Cropland net primary production (NPP) plays a crucial role in both food security
and atmospheric $CO_2$ variations. Crop yield is part of crop NPP, thus food security
relies greatly on crop NPP. It has been reported that increase in cropland NPP driven
by the agricultural Green Revolution contributed 17-45 % of the increase in
atmospheric $CO_2$ seasonal amplitude (Gray et al., 2014; Zeng et al., 2014).
Furthermore, vegetation is the most active C reservoir in the terrestrial ecosystem, and
is easily affected by climate change (e.g., drought) and management practices, thus
potentially affecting global climate change (Le Quéré et al., 2016; Zeng et al., 2005b;
Zhao and Running, 2010).
Globally, agricultural areas cover ~1,370 million hectares (Mha), distributed across
diverse climatic and edaphic conditions, with a variety of complex cropping systems
and management practices (Foley et al., 2011; Gray et al., 2014; Lal, 2004; Monfreda
et al., 2008). Features of the agricultural Green Revolution include 1) adoption of
improved varieties, 2) expansion of irrigation, and 3) increased use of chemical
fertilizer and pesticide. These three factors have contributed approximately equally to
increased crop NPP (Sinclair, 1998). Although the agricultural Green Revolution has
been identified as a key driver of increased crop yield, its impact on crop NPP differs
across time and space. Management intensity (here, mainly referring to the third
feature of the Green Revolution) varies largely and has not always changed
synchronously in different parts of the world (Table 1) (Ejeta, 2010; Evenson, 2005;
Glaeser, 2010; Hazell, 2009). Thus, cropland NPP is highly variable, complicating the
assessment of global cropland NPP (Bondeau et al., 2007; Ciais et al., 2007; Gray et
al., 2014). For example, in the USA, the timing and magnitude of the agricultural
Green Revolution occurred almost evenly from 1961-2010, while in Brazil, the most
dramatic increase occurred after 2000 (Glaeser, 2010; Hazell, 2009). However,
accounting for such effects of heterogeneity in management practices over time and
space on crop NPP at a global scale has been rare to date.



Three methods are available for estimating vegetation NPP: statistical data, process-based models and remote sensing. Statistical data and process-based models are prevalent method for estimating global NPP, but, except for a few recent studies, are generally limited to natural vegetation based on climate and edaphic variables, (Gray et al., 2014; Zeng et al., 2014). Therefore, global- and regional-scale estimates of cropland NPP therefore must rely on census and survey data. However, these data report agricultural production, not NPP, and thus need crop-specific factors (dry matter fraction, harvest index, root to shoot ratio, etc.) to calculate the NPP (Gray et al., 2014; Huang et al., 2007; Monfreda et al., 2008; Prince et al., 2001), which neglected the temporal evolution for crop-specific factors such as harvest index and root to shoot ratio (Lorenz et al., 2010; Sinclair, 1998). Remote sensing by satellites is a powerful tool for estimating global terrestrial NPP (Cleveland et al., 2015; Field et al., 1995; Nemani et al., 2003; Parazoo et al., 2014; Zhao and Running, 2010), yet croplands are coincident with natural vegetation, making it difficult to differentiate the two using remote sensing (Defries et al., 2000; Monfreda et al., 2008).

The current state of the global carbon models is as follows: 1) some models, such as LPJ or ORCHIDEE, do not have an agricultural module; 2) models with an agricultural module, such as LPJ managed Land (LPJmL), do not fully represent the features of the Green Revolution; 3) the VEGAS model, by Zeng et al. (2014), was the first attempt to model the agricultural Green Revolution. The importance of parameter calibration has been recognized and addressed by numerous modelling studies (Bondeau et al., 2007; Chen et al., 2011; Crowther et al., 2016; Luo et al., 2016; Ogle et al., 2010; Peng et al., 2013). In addition, regional calibrated parameters are critical for global-scale modelling (Le Quéré et al., 2016). However, because the management data needed for most terrestrial models is spatially and temporally scarce, a precise regional simulation and calibration seems impossible (Bondeau et al., 2007).

Here, we conducted a study concentrated on calibrations at both the regional and the country scale. Instead of using an extensive set of actual management data that are unavailable or incomplete, we modelled the first-order effects on crop NPP using parameterizations. Our objectives were to 1) describe the method for simulating the



three Green Revolution features, 2) quantify the cropland NPP over the last fifty years
on both the continental and country scales, and 3) improve the model's performance
by key parameterization.

## 2 Materials and methods

### 2.1 Simulating the Green Revolution with a dynamic vegetation model

We simulated agriculture using a generic crop functional type that represents an
average of three dominant crops: maize, wheat and rice. These crops are similar to
warm C3 grass, one of the natural plant functional types in VEGAS (Zeng et al.,
2005a; Zeng et al., 2014). A major difference is the narrower temperature growth
function, to represent a warmer temperature requirement than natural vegetation.
Cropland management is modelled as an enhanced photosynthetic rate by the cultivar
selection, irrigation and application of fertilizers and pesticides. We modelled the
first-order effects on carbon cycle using regional-scale parameterizations with the
following rules.

### 2.1.1 Variety

The selection of high-yield dwarf crop varieties has been a key feature of the
agricultural Green Revolution since the 1960s, generally accompanied by an increase
in the harvest index (the ratio of grain to aboveground biomass) (Sinclair, 1998). The
harvest index (HI) varies for different crops, with a lower value for wheat (0.37-0.43)
(Huang et al., 2007; Prince et al., 2001; Soltani et al., 2004) and higher values for rice
(0.42-0.47) (Prasad et al., 2006; Witt et al., 1999) and maize (0.44-0.53) (Huang et al.,
2007; Prince et al., 2001). We used a value of 0.45 for the year 2000, a typical value
of the three major crops: maize, rice and wheat (Haberl et al., 2007; Sinclair, 1998).
The temporal change of HI is modelled as:
$$HI_{crop} = 0.45(1 + 0.6\tanh(\frac{y-2000}{70}))$$ (1)
so that HI$_{crop}$ was 0.31 at the beginning of the Green Revolution in 1961, and 0.45 for



2000 (Fig. 1), based on values found in the literature (Prince et al., 2001; Sinclair,

111   1998).

### 2.1.2 Irrigation

To represent the effect of irrigation, the soil moisture function ($\beta = w_1$ for
unmanaged grass, where $w_1$ is surface soil wetness) is modified as:
$$\beta = 1 - \frac{(1-w_1)}{W_{irrg}} \tag{2}$$
The irrigation intensity $W_{irrg}$ varies spatially from 1 (no irrigation) to 1.5 (high
irrigation), with $\beta$ ranging from 0 (no irrigation) to 0.33 (high irrigation) under
extreme dry natural conditions (Fig. 2). This function also modifies $\beta$ when $w_1$ is not
zero, but the effect of irrigation decreases when $w_1$ increases and levels off when $w_1$
equals 1 (soil is saturated). Thus, $\beta$ (and thus the photosynthesis rate) is determined by
both naturally available water ($w_1$) and irrigation. The spatial variation in $W_{irrg}$ reflects
a regional difference between tropical and temperate climates.

### 2.1.3 Fertilizer and pesticide

To represent the enhanced productivity from cultivar and fertilization, the gross
carbon assimilation rate is modified by a management intensity factor (MI) that varies
spatially and changes over time:
$$MI(region, year) = M_0\, M_1(region, MAT(lat, lon))\, M_2(year) \tag{3}$$
$$M_1(region, MAT) = M_{1r}(region) * Max(1 - tanh(MAT(lat, lon) - 15/25), 1.0)$$
$$\tag{4}$$
$$M_2(year) = 1 + 0.2 tanh(\frac{year - 2000}{70}) \tag{5}$$
where $M_0$ is a scaling factor, the default value taken as 1.7 compared with natural
vegetation 1.0, while $M_1$ is the spatially varying parameter, using major global regions
as listed in Table 2 and mean annual temperature (MAT) to differentiate (Eq. 2). $M_{1r}$ is
a region-dependent relative management intensity factor and $M_1$ is stronger in
temperate and cold regions and weaker in tropical countries, for which we used the
mean annual temperature as a surrogate (Eq. 2). $M_2$ is a temporal evolutionary factor



(Eq. 3), and the term in parentheses represents the temporal evolution, modelled by a
hyperbolic tangent function, with the MI values in 1961 approximately 10 % lower
than in 2000, and 20 % lower asymptotically farther back in time (Fig. 3).

### 2.1.4 Motivation of the $M_{1r}$ parameter calibration

$M_{1r}$ is a region-dependent relative management intensity factor that varied largely
across regions, and the default parameters were derived from a previous version used
in Zeng et al. (2014), mainly to capture the global trends, which neglected the
regional trends to some degree. A main focus of this study is to improve the $M_{1r}$
parameter based on the FAO regional data to capture the regional trends. For each
individual region, we used a series of parameters to drive the model and chose the
best fit for the FAO statistical data (by naked eye observation) as follows:
1. Parameter $M_{1r}$ was calibrated on a continental scale to match the FAO statistical
data. During this period, countries within the same continent were assigned the
same $M_{1r}$.
2. The $M_{1r}$ for selected major countries was calibrated independently from the
continental calibration, while the other countries that were not selected within the
same continent were tuned oppositely from the selected countries to keep the total
simulated continental production close to the FAO data.
After the two steps, total production was summed as all countries with updated
parameters.

### 2.1.5 Planting, harvesting, and lateral transport

Crop phenology was not decided beforehand but was determined by the climate
condition. For example, when it is sufficiently warm in temperate and cool regions,
crops begin to grow. This assumption captures most of the spring planting, and
simulates multiple cropping in low latitudes. However, one limitation of such simple
assumption is that it misses some other crop types, such as winter wheat, which has an
earlier growth and harvest.



When the leaf area index (LAI) growth rate slows to a threshold value, a crop is
assumed to be mature and is harvested. The automatical planting and harvest criteria
allow multiple cropping in some warm regions, and matches areas with intense
agriculture such as East Asia and Southeast Asia, but the criteria may overestimate
regions with single cropping. Consequently, the simulated results tend to be the
potential productivity due to the climate characteristics and our generic crop.
After harvest, grain and straw are assumed to be appropriated by farmers and then
incorporated into the soil metabolic carbon pool. The harvested crop is redistributed
according to population density, resulting in the horizontal transport of carbon. As a
consequence, cropland areas act as net carbon sinks, and urban areas release large
amount of $CO_2$ through heterotrophic respiration. Lateral transport is applied within
each continent to simulate the first-order approximation. Additional information on
cross-regional trade was also taken into account for eight major world economic
regions.
**2.2 Data sets**
**Climate data**
Gridded monthly climate data sets (i.e., maximum and minimum temperature,
precipitation, and radiation) covering the period 1901–2013 with a spatial resolution
of 0.5 °×0.5 ° were obtained from the Climatic Research Unit, University of East
Anglia (http://www.cru.uea.ac.uk/cru/data/hrg/). The CRU TS3.22 (Harris et al., 2013)
are calculated on high-resolution grids, which are based on an archive of monthly
mean temperatures provided by more than 4000 weather stations distributed around
the world. The dataset has been widely used for global change studies (Mitchell et al.,
2004; Mitchell and Jones, 2005).
**Land-cover data**
The land-cover data set (crop/pasture versus natural vegetation) was derived from
the History Database of the Global Environment (HYDE) data set
(http://themasites.pbl.nl/tridion/en/themasites/hyde/download/index-2.html)





(Goldewijk et al., 2010; Goldewijk et al., 2011). It is an update of the HYDE with
estimates of some of the underlying demographic and agricultural driving factors
using historical population, cropland and pasture statistics combined with satellite
information and specific allocation algorithms. The 3.1 version has a 5′
longitude/latitude grid resolution, and covers the period 10,000 BC to AD 2000. This
data set was also used in TRENDY and other model comparison projects (Chang et al.,
2017; Sitch et al., 2015). The VEGAS model does not use high spatial resolution
land-use and management data such as crop type and harvest practices; thus,
small-scale regional patterns may not be well simulated, and the results are more
reliable at aggregated continental to global scales.
**Crop production data**
Crop production and cropland area are aggregated from FAO statistics for the major
crops (FAOSTAT, http://www.fao.org/faostat/en/#data/QC, accessed June 2016).
Specifically, they are the sum of the cereals (wheat, maize, rice, and barley, etc.) and
five other major crops (cassava, oil palm, potatoes, soybean and sugar-cane), which
comprise 90 % of the global amount of carbon harvested. Following Ciais et al.
(2007), conversion factors are used to convert first wet to dry biomass, then to carbon
content. The final conversion factors from wet biomass to carbon are 0.41for cereals,
0.57 for oil palm, 0.11 for potatoes, 0.08 for sugarcane, and 0.41 for soybean and
cassava.
**2.3 Initialization and simulation**
The VEGAS model used in TRENDY (Sitch et al., 2015; Zeng et al., 2005a) was
run from 1700 to 2010 and, forced by climate, annual mean $CO_2$, and land-use and
management history. Due to unavailable data of observed climate data before 1900,
the average climate data over the period from 1900 to 1909 was used to drive the
"spin-up". The VEGAS model has a speed up procedure for soil carbon to make it
achieve equilibrium state (Zeng et al., 2005).



## 3 Results

### 3.1 A brief revisit of the agricultural Green Revolution

The agricultural Green Revolution was mostly started in the 1960s to cope with the food–population balance, particularly in developing countries (Borlaug, 2002) (Table 1). Its features include the development of high-yield varieties (HYVs) of cereal grains, the expansion of irrigation, and applications of synthetic fertilizers and pesticides (Borlaug, 2007). The intensity of such management varies widely and has not always occurred synchronously in different parts of the world. Specifically, in the 1950s, new wheat and maize varieties were developed by the International Maize and Wheat Improvement Center (CIMMYT) in Mexico, and their agricultural productivity increased with irrigated cultivation in the northwest (Byerlee and Moya, 1993; Gollin, 2006; Pingali, 2012). Later in 1966, a new dwarf high-yield rice cultivar, IR8 was bred by the International Rice Research Institute (IRRI) in the Philippines, and it was spread and grown in most of the rice-growing countries of Asia, Africa and Latin America (Fischer et al., 1998; Khush, 2001; Peng et al., 1999). Also in the 1960s, India imported new wheat seed from CIMMYT to Punjab and later adopted IR8 rice variety from Philippines that could produce more grains (Parayil, 1992). China began participating in the Green Revolution in the 1970s, with hybrid rice bred by Longping Yuan (Yuan, 1966), and the fertilizer application rate increased dramatically from 43 kg/ha in 1970 to 346 kg/ha in 1995 (Hazell, 2009). Meanwhile, Brazil began participating in the Green Revolution in the 1970s, and in collaboration with CIMMYT, high-yielding wheat varieties with aluminum toxicity resistance were developed, which were efficient in dealing with the aluminum toxicity in the Cerrado soils of Brazil (Davies, 2003; Khush, 2001). In contrast, African countries began their participation in the Green Revolution much later in the 1980s, with many obstacles from both climatic, edaphic and social-economic factors (Ejeta, 2010; Sánchez, 2010) and it featuring sustainable agriculture, plant breeding, and biotechnology.



## 3.2 Global and continental comparison between model simulation and FAO statistical data

Worldwide, the FAO data showed that cropland production increased from 439 TgC in 1961 to 1519 TgC in 2010 (246 % increase) (Fig. 4), and the VEGAS model captured most of this trend in both the default and the calibrated results. East Asia and North America contributed the most to this trend (Fig. 5). For East Asia, crop production increased from 65 TgC in 1961 to 342 TgC (426 % increase) in 2010. For North America, it increased from 90 TgC in 1961 to 235 TgC (161 % increase) in 2010. Other regions followed the increasing trend except for the former USSR region. The lowest crop production existed in Central-West Asia and Oceania, with less than 50 TgC over the study period.

As described in Sect. 2.1.4, we calibrated the $M_{1r}$ parameter for each region. The default and updated regional management intensity parameter (Table 2) produced dramatically different estimations for some continents, for example in North America, Southeast Asia and Oceania (Fig. 5b, e, j). However, for other continents, such as South Asia, the improvement was not so pronounced. For East Asia, the default parameter was sufficient to capture most of the crop production variations. Moreover, the timing and magnitude of the agricultural Green Revolution was quite different over different regions. For example, it occurred more recently in Africa and South America (Fig. 5a,c) and much earlier in East Asia and Europe (Fig. 5d, i). In the region of former USSR, crop production even decreased after 1990 (Fig. 5h) due to the large areas of abandoned croplands, thus making the regional-scale simulation more complicated.

Furthermore, the updated parameters in different regions did not substantially change the total production estimations (Fig. 4), indicating that a good agreement in global total production may be overestimated in some regions while underestimated in others, which does not reflect the true nature of the production distributions and variations.





**3.3 Country-scale comparison between model simulation and FAO statistical data**

At the country level, the FAO data showed that China, the USA and India were the top three countries contributing to global crop production (Fig. 6). For China, crop production increased from 50 TgC in 1961 to 230 TgC in 2010 (360 % increase). For the USA, it increased from 76 TgC in 1961 to 204 TgC in 2010 (168 % increase). Other countries followed the same increasing trend with different rates. The lowest crop production in the top 9 countries existed in Canada and Argentina, with less than 50 TgC over the study period.

As for the VEGAS simulations, the default parameters (Table 3) might overestimate results in some countries while underestimating others. The calibrated parameter could capture variations in most of the countries (Fig. 6). For Chinese crop production, a decreasing trend after 1999 was captured, but the magnitude was weaker (Fig. 6a), because the drop in cropland area was not represented in HYDE 3.0 for China. The calibrated parameter also performed well in other countries. For Brazil and Argentina, the dramatic increase after 2000 was not well captured due to the simple assumption that the strongest management occurred in 2000 and became weaker afterwards.

Based on the country-scale comparisons between the updated VEGAS simulations and the FAO statistical data of the decadal means, the linear regression slope was 1.00, with a higher $R^2$ of 0.97 (p < 0.01), a smaller RMSE of 10.5 TgC (~45 % decrease), and a smaller RMD of 3.5 TgC (~31 % decrease) compared with the default results (Fig. 7).

**3.4 Spatial comparison between the model simulation and the documented data**

The two independent datasets produced similar spatial distributions of crop NPP (Fig. 8). The highest crop NPP regions were the Great Plains of North America and temperate western Europe and East Asia (> 1.0 Tg per 0.5 °grid cell, Fig. 8), where the agricultural Green Revolution was the strongest, but high yields were also present





locally within tropical regions (e.g., Southeast Asia), while the lowest production in
Africa, Eastern Europe and Russia (< 0.4 Tg per 0.5 °grid cell, Fig. 8) was due largely
to the low input in agricultural R & D and the rigid climate and edaphic conditions.
The model result overestimated Russian cropland NPP because of the simplified
model representation of temporal changes, and the abandoned cropland after the
collapse of former USSR was not represented in the HYDE data set. Meanwhile, the
high South American NPP was underestimated.
The average cereal NPP increased from 1.0 Mg ha$^{-1}$ to 1.5 Mg ha$^{-1}$ for African
croplands (Fig. 9a), and it increased from 1.5 to 2.1 Mg ha$^{-1}$ for Oceania croplands
from 1961 to 2014. Europe, Asia and South America showed similar increasing trends
from 1.5 to 4.0 Mg ha$^{-1}$. North America showed the highest cereal NPP, with an
increase of 2.5 to 8.0 Mg ha$^{-1}$ over the fifty years. For soybean NPP, America topped
the six continents with 3.0 Mg ha$^{-1}$ in 2010, while Africa showed the lowest NPP with
1.2 Mg ha$^{-1}$ in 2010, one-third that of America. Europe and Oceania had a middle
level of ~2.0 Mg ha$^{-1}$ in 2010. This NPP trend was consistent with the progress of the
Green Revolution progress on each continent.

## 4   Discussion

In the estimation of crop NPP, one of the sources of uncertainty is crop parameters,
such as variations in harvest index. When accounting for this variation of 0.45
(0.37-0.53, or 18 % of the mean), the uncertainty resulted from the harvest index for
the FAO production derived NPP would be 1.3 ± 0.2 and 3.6 ± 0.6 PgC yr$^{-1}$ in the
1960s and 2010s, respectively. Additionally, one of the main driving factors for the
agricultural Green Revolution was the economic input. Gross domestic expenditures
on food and agricultural R&D worldwide has increased from 27.4 to 65.5 billion of
2009 purchasing power parity (PPP) dollars from 1980-2010 (Pardey et al., 2016).
The middle-income countries R&D investment share increased from 29 % in 1980 to
43 % in 2011. This investment difference has dramatically influenced the crop NPP
(Fig. 4, 5, 6, 8) due to improvements in crop varieties, fertilizer and pesticide





application, and expansion of irrigation areas (Ejeta, 2010; Evenson, 2005; Evenson
and Gollin, 2003; Gollin et al., 2005; Gray et al., 2014; Hazell, 2009). Despite a
drought-induced reduction in the global terrestrial NPP of 0.55 PgC from 2000 to
2009 based on MODIS satellite data analysis (Zhao and Running, 2010), cropland
NPP increased 0.3-0.6 PgC for the same period in this study because of the
agricultural Green Revolution (Fig. 4).
Gray et al. (2014) used production statistics and a carbon accounting model to show
that increases in agricultural productivity explained ~25 % changes in atmospheric
$CO_2$ seasonality. Northern Hemisphere extratropical maize, wheat, rice, and soybean
production increased 0.33 PgC (240 %) between 1961 and 2008. This study showed a
consistent estimation: the total cropland production increased 1.0 PgC (300 %), and
took up 0.5 Pg more carbon in July. Furthermore, Monfreda et al. (2008) estimated the
global cropland NPP for the year 2000 at the subcountry scale using the FAO
statistical yield data and cropland area distributions. Consistently, the global cropland
mean NPP was estimated as 4.2 MgC ha$^{-1}$, with the highest NPP in Asian croplands of
5.5 MgC ha$^{-1}$ and the lowest in African croplands of 2.5 MgC ha$^{-1}$. Specifically, both
studies agreed well in several regions that had the highest cultivated NPP due to
intensive agriculture and/or multiple cropping: Western Europe; East Asia; the central
United States; and southern Brazil, with NPP larger than 10 MgC ha$^{-1}$. Meanwhile,
Bondeau et al. (2007) modelled the difference of agricultural NPP between LPJmL
and LPJ, showing that agriculture increased NPP in intensively managed or irrigated
areas (Europe, China, southern United States, Argentina). However, their study could
not capture the increasing trends in the US Central Plains and in the Australian wheat
belt because of the unavailability of management data at those regional scales,
showing the limitations of modelling using detailed regional management data.
Moreover, using country-based agricultural statistics and activity maps of human and
housed animal population densities, Ciais et al. (2007) estimated the global carbon
harvested in croplands was 1.3 PgC yr$^{-1}$, of which ~13 % enters into horizontal
displacement through international trade circuits, contributing ~0.2-0.5 ppm mean
latitudinal $CO_2$ gradients.



European cropland NPP increased 127 % over the last half century, as estimated by
VEGAS (Fig. 5i), and the yield increased at a rate of 1.8 % per annum. Moreover,
without the management intensity parameter updated, the crop yields for the 2000s
would be 10.4 % lower. Similarly, a study showed that across all major crops
cultivated in the EU, plant breeding has contributed approximately 74 % of total
productivity growth since 2000, equivalent to a yield increase of 1.2 % per annum.
European crop yields today would be more than 16 % lower without access to
improved varieties (BSPB). The 2003 drought and heat in Europe reduced the
terrestrial gross primary productivity (GPP) by 30 % (Ciais et al., 2005), while it was
decreased by 15 % for cropland NPP in this study (Fig. 5i). This decrease was smaller
than the natural ecosystem response due largely to the counteractive effects of
management inputs (irrigation, fertilization, etc.).
In the central USA, VEGAS modelled the cropland NPP as > 6 MgC ha$^{-1}$ in the
Great Plains and < 3 MgC ha$^{-1}$ in northwest and north USA for the 2000s. Prince et al.
(2001) estimated crop NPP by applying crop-specific factors to statistical agricultural
production. The NPP at the county-level in 1992 ranged from 2 MgC ha$^{-1}$ in North
Dakota, Wisconsin, and Minnesota to >8 MgC ha$^{-1}$ in central Iowa, Illinois, and Ohio.
Areas of highest NPP were dominated by corn and soybean cultivation. Using a
similar method, Hicke et al. (2004) estimated crop NPP increased in counties
throughout the United States, with the largest increases occurring in the Midwest,
Great Plains, and Mississippi River Valley regions. It was estimated that total
coterminous cropland production increased from 0.37 to 0.53 (a 40 % increase) Pg C
yr$^{-1}$ during 1972–2001.
In Asian croplands, the percentage of harvested area for rice, wheat and maize
under modern varieties was lower than 10 % in the 1960s, and it increased to over 80 %
in the 2000s (Evenson, 2005). Moreover, nitrogen (N) fertilizer increased from 23.9
kg ha$^{-1}$ in 1970 to 168.6 kg ha$^{-1}$ in 2012, while the irrigated area increased from 25.2 %
in 1970 to 33.2 % in 1995 (Rosegrant and Hazell, 2000). Correspondingly, the crop
NPP increased from 1.4 in 1961 to 4.5 MgC ha$^{-1}$ in 2014 (Fig. 9). Cropland NPP in
China was estimated to increase from 159 TgC yr$^{-1}$ in the 1960s to 513 TgC yr$^{-1}$ in the



1990s based on the National Agriculture Database (Statistics Bureau of China 2000) (Huang et al., 2007), and this study estimated the range as 286 TgC yr$^{-1}$ in the 1960s to 559 TgC yr$^{-1}$ in the 1990s. In tropical Asia, the new croplands were mainly derived from forests, which caused large amounts of carbon losses from both vegetation and soil (Gibbs et al., 2010; Tao et al., 2013; West et al., 2010).

The African croplands currently nourish over 1.0 billion people. The need for sustainable agriculture combined with stable grain yield production is particularly urgent in Africa. However, the continent is now trading carbon for food. Newly cleared land in the tropics releases nearly 3 tons of carbon for every 1 ton of annual crop yield compared with a similar area cleared in the temperate zone (West et al., 2010). This continent can triple its crop yields provided the depletion of soil nutrients is addressed (Sánchez, 2010). Using chemical fertilizer as an example, the average N application rate from 2002 to 2012 was only ~14 kg ha$^{-1}$ yr$^{-1}$ in Africa, which severely hampered crop production (Han et al., 2016). In addition, complete crop residue removal for fodder and fuel is a norm in Africa, causing soils in these areas to lack organic matter input and to become carbon sources (Lal, 2004). Since the mid-1970s, ~50 Mha of Ethiopian land had no or low fertilizer application, resulting in low crop NPP (< 2 MgC ha$^{-1}$, Fig. 7, 8) (West et al., 2010) and soil degradation (Shiferaw et al., 2013). African agricultural development has to overcome a series of constraints such as drought, poor soil fertility, diverse agro-ecologies, unique pests and diseases, and persistent institutional and programmatic challenges (Ejeta, 2010).

In terms of the data gap in management intensity, very few data sets provide long-term time series data with high spatial resolution. HYDE is a land use dataset that does not provide management intensity information (Goldewijk et al., 2011). Monfreda et al. (2008) developed a data set consisting of 175 crops consistent to the FAO statistical data for the period around year 2000. Moreover, Fritz et al. (2015) developed a cropland percentage map for the baseline year 2005. For the fertilizer dataset, Potter et al. (2010) provided the global manure N and P application rate for a mean state around year 2000. Moreover, Lu and Tian (2017) developed a global time series gridded data set for synthetic N and phosphorous (P) fertilizer application rate





in agricultural lands. For the irrigation data set, global monthly irrigated crop areas
around the year 2000 were developed by Portmann et al. (2010). These data sets are
mostly for a specific year or a period mean, and they are unsuitable for long-term
simulations. Therefore, we still lack a comprehensive data set that reflects
management intensity.
A more challenging task would be to calibrate regional parameters and explain
spatial patterns better, because models may significantly underestimate the
high-latitude trend (Graven et al., 2013) and overestimate elsewhere even if the global
total is simulated correctly (Zeng et al., 2014). More work should be directed to
reduce uncertainties in regional model parameterizations (Le Quéré et al., 2015; Luo
et al., 2016). This paper focuses on both the continental and country scales to calibrate
key parameters to better constrain the future projections of global cropland NPP.

## 5   Conclusion

We used a process-based terrestrial model VEGAS to simulate global cropland
production from 1960 to 2010, and adapted the management intensity parameter at
both continental and country scales. The updated parameter could capture the
temporal dynamics of crop NPP much better than the default ones. The results showed
that cropland NPP tripled from $1.3 \pm 0.1$ in the 1960s to $3.6 \pm 0.2$ Pg C yr$^{-1}$ in the
2000s. The NPP increased most notably in the U.S. Midwest, Western Europe and the
North China Plain. In contrast, it increased slowly in Africa and Oceania. We
highlight the large difference in model parameterization among regions when
simulating the crop NPP due to the differences in timing and magnitude of the Green
Revolution. To better explain the history and predict the future crop NPP trends, it is
important to calibrate key parameters at regional scales and develop time series data
sets for land management history.





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

**Tables:**

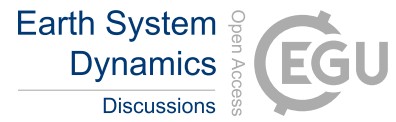

Table 1 Features of the agricultural Green Revolution across regions

| Region/Country | Starting period | Features | Ref. |
|---|---|---|---|
| Africa | 1980s | Sustainable agriculture, plant breeding, and biotechnology | (Evenson and Gollin, 2003); (Ejeta, 2010);(Pingali, 2012) |
| Asia | 1960s | Varieties breeding, use of chemical fertilizers and pesticides, and irrigation | (Hazell, 2009) |
| Europe and North America | 1960s | Large public investment in crop genetic improvement built on the scientific advances for the major staple crops —wheat, rice, and maize | (Pingali, 2012) |
| South America | 1960s | Varieties breeding, use of chemical fertilizers and pesticides, and irrigation | (Evenson and Gollin, 2003); (Hazell, 2009) |
| Mexico | 1950s | New wheat and maize varieties developed by the International Maize and Wheat Improvement Center. Improve agricultural productivity with irrigated cultivation in northwest | (Cotter, 2005); (Khush, 2001);(Pingali, 2012) |
| Philippines | 1966 | A new dwarfed high-yield rice cultivar, IR8 was bred by IRRI | (Fischer et al., 1998); (Peng et al., 1999) |





| | | | |
|---|---|---|---|
| India | 1960s | Plant breeding, irrigation development, and financing of agrochemicals | (Hazell, 2009); |
| China | 1970s | Hybrid rice bred by Longping Yuan; Fertilizer increased dramatically | (Yuan, 1966); (Lin and Yuan, 1980) |
| Brazil | 1970s | High-yielding wheat varieties with aluminum toxicity resistance were developed | (Davies, 2003);(Khush, 2001);(Marris, 2005) |



Table 2 Default and calibrated regional management intensity parameter of $M_{1r}$. The
default values were obtained from Zeng et al., (2014), which were parameterized
mainly for global trend simulation. See Sect. 2.1.4 for the calibration. Updated $M_{1r}$
values are represented by ↑ and ↓ symbols, indicating an increase or a decrease
compared to the default ones, respectively.

| Continent | Default | Calibrated |
|---|---|---|
| Africa | 0.5 | 0.8↑ |
| North America | 1.3 | 1.1↓ |
| South America | 0.7 | 0.9↑ |
| East Asia | 1.5 | 1.5 |
| Southeast Asia | 1.0 | 0.7↓ |
| South Asia | 0.7 | 0.6↓ |
| Central-West Asia | 0.7 | 1.0↑ |
| Former USSR | 1.0 | 1.2↑ |
| Rest of Europe | 1.3 | 1.1↓ |
| Oceania | 1.0 | 0.6↓ |


Table 3 Default and calibrated national management intensity parameter of $M_{1r}$.

| Country | Default | Calibrated |
|---|---|---|
| China | 1.5 | 1.3↓ |
| USA | 1.3 | 1.0↓ |
| India | 0.7 | 0.6↓ |
| Russia | 1.0 | 0.9↓ |
| Brazil | 0.7 | 0.8↑ |
| Indonesia | 1.0 | 0.7↓ |
| France | 1.3 | 3.0↑ |
| Canada | 1.3 | 2.1↑ |
| Argentina | 0.7 | 0.8↑ |




**Figure Captions:**

Figure 1: Harvest index change over time as used in the model, and a harvest index of 0.31 in 1961 and 0.49 in 2010, based on literature review.

Figure 2: Irrigation intensity ($W_{irrig}$) changes with mean annual temperature (MAT) and β (beta) changes with soil wetness for typical $W_{irrig}$ as used in the model.

Figure 3: Management intensity (relative to year 2000) changes over time as used in the model. The analytical functions are hyperbolic tangent (see text). The parameter values correspond to a management intensity in 1961 that is 10 % smaller than in 2010.

Figure 4: Annual global crop production from 1961 to 2010. Default parameters were derived from a previous version that was used in Zeng et al., (2014) to capture the global trends, and calibrated parameters were set in this study (see text) to capture the regional trends.

Figure 5: Annual crop production from 1961 to 2010 at continental scales. The (d) subplot has no purple line since the default parameter produced the best fit for all the tuned simulations.

Figure 6: Annual crop production from 1961 to 2010 at country scales.

Figure 7: Country-based comparison of simulated and observed cropland productions (Tg) before (a) and after (b) calibration. Each country consists of five dots representing the five decadal mean values, respectively.

Figure 8: Mean cropland NPP from 1997 to 2003. VEGAS modelled patterns (in units of Tg C per 0.5 °grid cell, upper panel) show major productions in the agricultural areas of North America, Europe and Asia (the lower panel shows the mean crop NPP based on the FAO statistical data from Navin Ramankutty (http://www.earthstat.org/).

Figure 9: Cereal and soybean NPP at continental scales over the last 60 years derived from FAO yield data. Note that the scales are different.





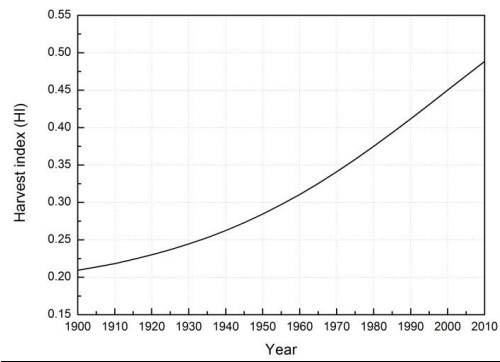


Figure 1: Harvest index change over time as used in the model, and a harvest index of
0.31 in 1961 and 0.49 in 2010, based on literature review.


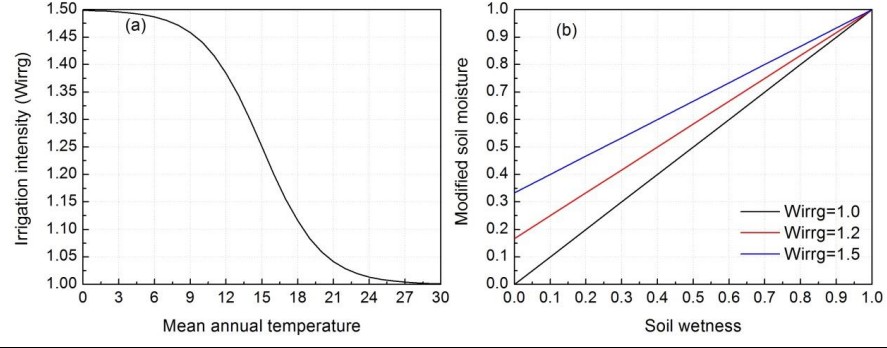


Figure 2: Irrigation intensity ($W_{irrig}$) changes with mean annual temperature (MAT)
and β (beta) changes with soil wetness for typical $W_{irrig}$ as used in the model.






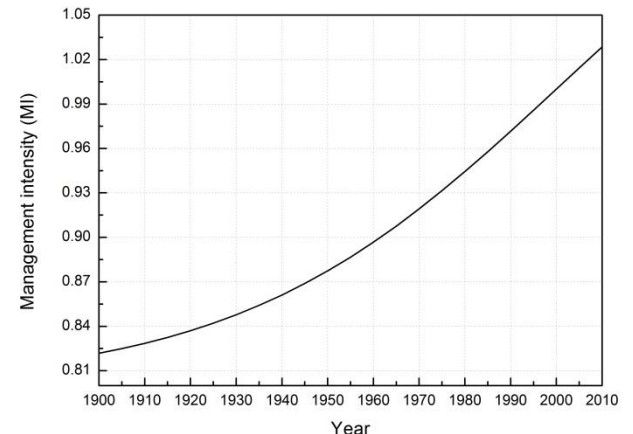


Figure 3: Management intensity (relative to year 2000) changes over time as used in
the model. The analytical functions are hyperbolic tangent (see text). The parameter
values correspond to a management intensity in 1961 that is 10 % smaller than in
2010.

699

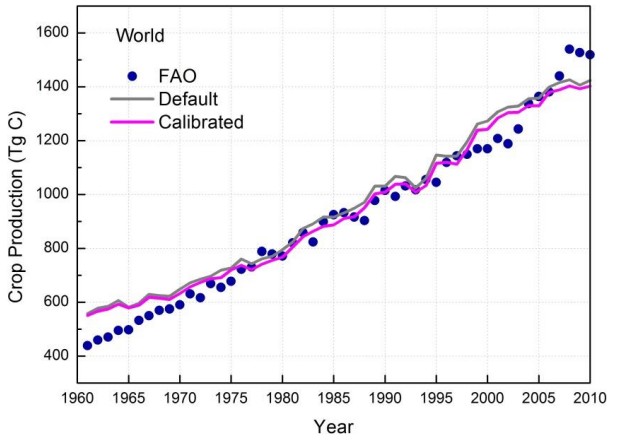

700

Figure 4: Annual global crop production from 1961 to 2010. Default parameters were
derived from a previous version that was used in Zeng et al., (2014) to capture the
global trends, and calibrated parameters were set in this study (see text) to capture the
regional trends.





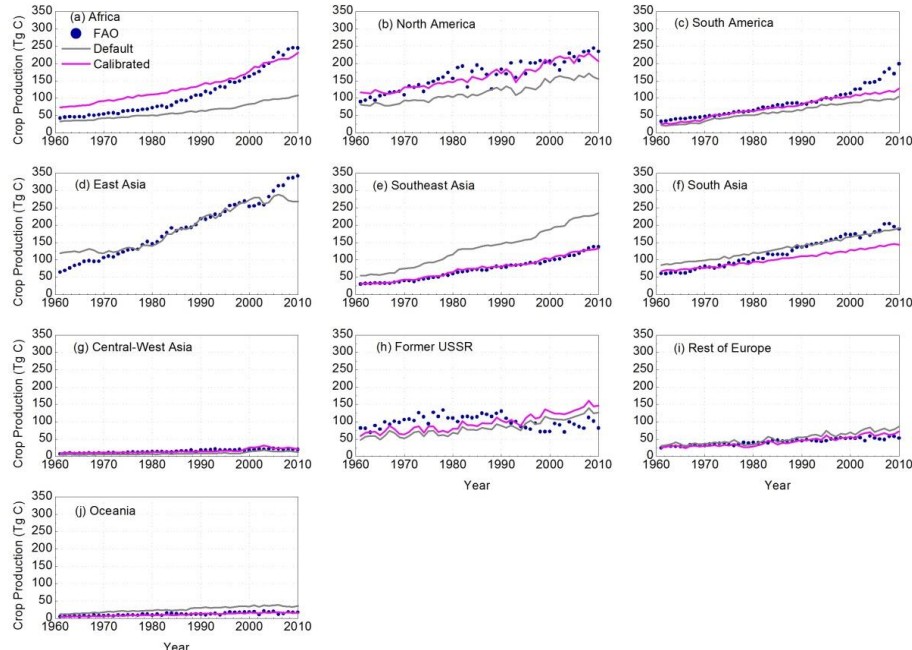


Figure 5: Annual crop production from 1961 to 2010 at continental scales. The (d)
subplot has no purple line since the default parameter produced the best fit for all the
tuned simulations.




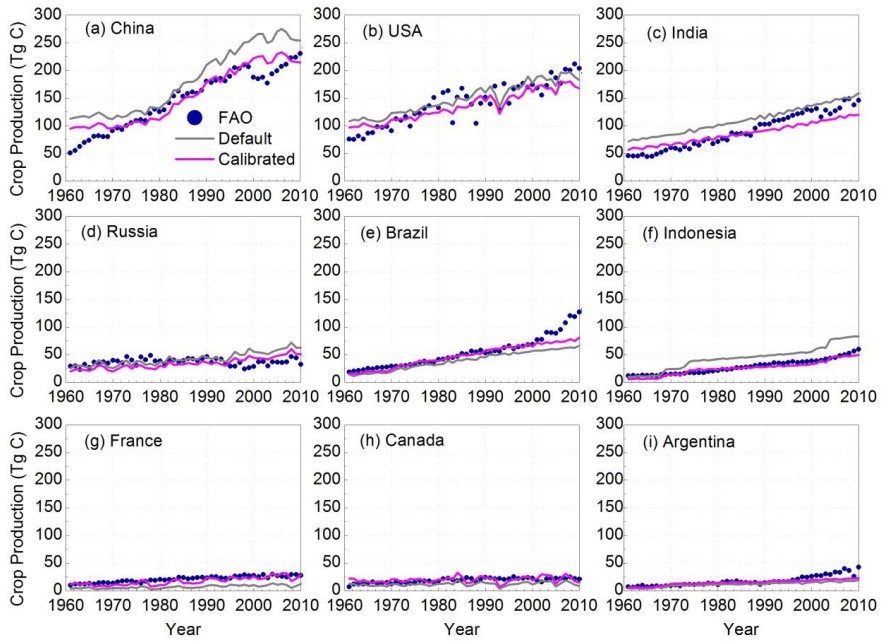

Figure 6: Annual crop production from 1961 to 2010 at country scales.

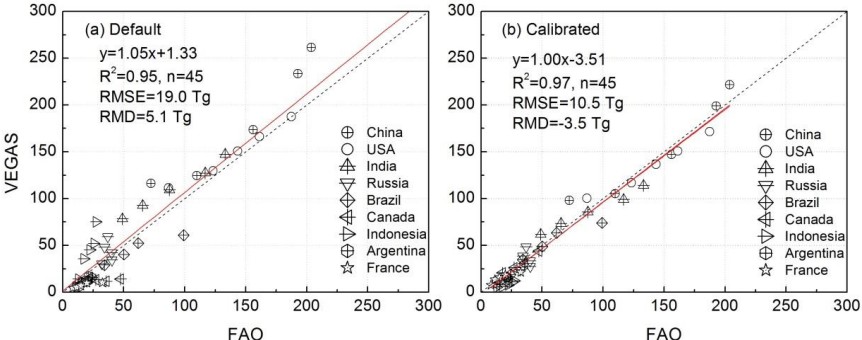

Figure 7: Country-based comparison of simulated and observed cropland productions
(Tg) before (a) and after (b) calibration. Each country consists of five dots
representing the five decadal mean values, respectively.



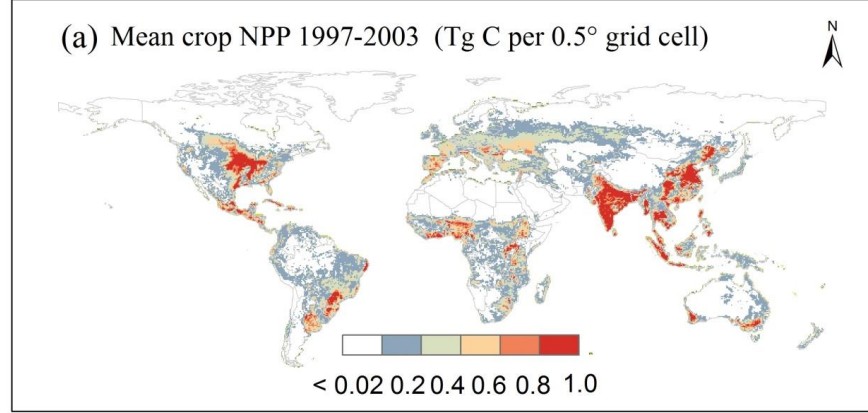

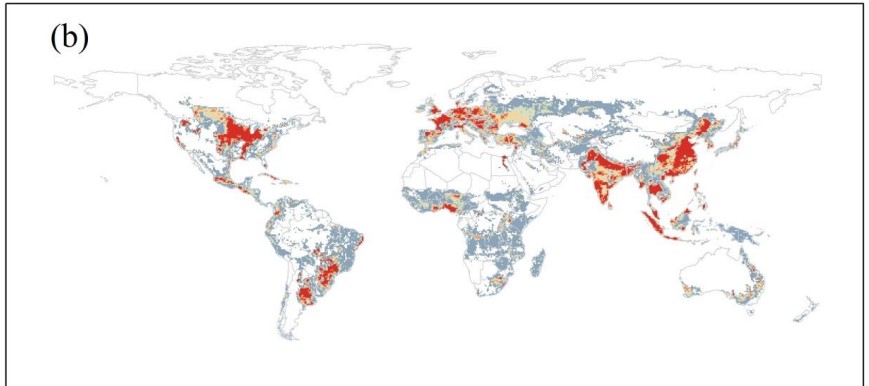


Figure 8: Mean cropland NPP from 1997 to 2003. VEGAS modelled patterns (in units
of Tg C per 0.5°grid cell, upper panel) show major productions in the agricultural
areas of North America, Europe and Asia (the lower panel shows the mean crop NPP
based on the FAO statistical data from Navin Ramankutty (http://www.earthstat.org/).






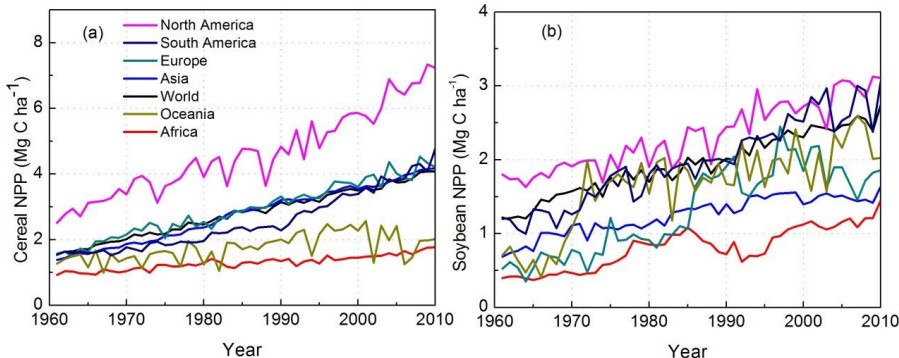


Figure 9: Cereal and soybean NPP at continental scales over the last 60 years derived
from FAO yield data. Note that the scales are different.
