# Peer review of "Estimating global cropland production from 1961 to 2010"

_Earth System Dynamics, 2017_

## Referee Comment (RC1) · Anonymous Referee #1 · 20 Jul 2017

This paper presents a study of global cropland production modeling using a process-based model VEGAS. The Green Revolution is addressed here which is very important in global carbon cycle. However, how to represent the Green Revolution in models is a major task. A prior attempt was made in VEGAS. Although large uncertainties and lots of future improvements remain, the results are still reasonable. Therefore, this paper fits the journal scope well and has certain scientific contributions.

There are only some issues to be addressed in the discussion section about the uncertainties.

1, The Green Revolution has not changed synchronously across different regions globally. However, the harvest index represented in equation 1 treats it identical over the globe. Please discuss the uncertainties of this issue.

2, Irrigation in section 2.1.2. As mentioned in the context, the irrigation intensity varies spatially. Please specify the sources of this spatial information? Is the spatial information from the HYDE data or generated by the model?

3, section 2.1.5. it says "Crop phenology was not decided beforehand but was determined by the climate condition." Double cropping over the East Asia is very popular where the climate conditions are sufficient. However, in the USA, single cropping is major under similar climate condition. Please discuss this issue a little bit in the discussion section.

4, DO double check the context and the references. There are too many small mistakes in the reference list, such as Subscript 2 in CO2. Mistakes were found in Lines 508 and 518 536. . . .. . . DO keep the references in an identical style.

---

## Referee Comment (RC2) · Anonymous Referee #2 · 28 Jul 2017

This study simulates global cropland NPP from 1961 to 2010 using the VEGAS model and compares the simulation with FAO statistical data on continental and country scales. The comparison indicates general agreement between the model simulation and the statistical data, yet the scientific importance of such comparisons may be questionable. Because the study essentially tuned the model parameters to fit the FAO data, the agreement found in the comparison may only demonstrate the success of the adopted model-tuning methods.

Some of the technical drawbacks of the studies include: 1) The temporal trend of the Green Revolution seems to be totally decided by Eqs. (1) and (5). Are the two equations (which use the same reference year 1960 and the same temporal scale factor of 70 years) representative for all continents/countries under consideration?

[Figure]

2) How should we interpret/compare the values of M1r? The values in Tables 2 and 3 appear as "magic numbers" to me. For instance, in Table 3 the new M1r values of France and Canada are roughly three times and twice as high as US. What does these values really mean?

3) Fig. 8 and corresponding text: Avoid the use of "Tg C per 0.5 deg grid cell" as the unit for crop NPP as the area of "0.5 deg grid cell" varies at different latitudes. The results shown in the figure thus are potentially misleading.

4) a minor comment: Lines 257-260 on Page 11 state that the adjusted M1r parameters produced "dramatically" different estimations for continents include Oceania (Fig. 5j). Why? It appears to me that the difference in Africa (Fig. 5a) is much more "dramatic" than Oceania.

Otherwise the paper is well written and easy to read.

―――――――――――――――――――

---

## Author Comment (AC1) · 19 Aug 2017

Reviewer 1: This paper presents a study of global cropland production modeling using a process-based model VEGAS. The Green Revolution is addressed here which is very important in global carbon cycle. However, how to represent the Green Revolution in models is a major task. A prior attempt was made in VEGAS. Although large uncertainties and lots of future improvements remain, the results are still reasonable. Therefore, this paper fits the journal scope well and has certain scientific contributions.

Response: Thanks for the understanding of this paper.

There are only some issues to be addressed in the discussion section about the uncertainties. 1, The Green Revolution has not changed synchronously across different

regions globally. However, the harvest index represented in equation 1 treats it identical over the globe. Please discuss the uncertainties of this issue.

Response: Thanks for the valuable advice. Indeed, the harvest index (HI) represented in equation 1 did not change over different areas. This was mainly constrained by the limited large scale observed values over time. And we mainly modeled the first order of the Green Revolution, which is the decreased HI trend over time. Furthermore, we discussed this uncertainty in Line 323-327.

2, Irrigation in section 2.1.2. As mentioned in the context, the irrigation intensity varies spatially. Please specify the sources of this spatial information? Is the spatial information from the HYDE data or generated by the model?

Response: Thanks for the question. We added the Eq. and description in the revised manuscript in Line 115 and 120. Irrigation data was generated by the VEGAS model using an empirical function related to mean annual temperature (MAT) (Fig 1):

Irrigation intensity = 1+0.5*(1/(1+EXP(2*(MAT-15)/5))).

We acknowledge this is a rough assumption due to the lack of global irrigation data set with time series. The irrigation intensity varies spatially from 1 (no irrigation) to 1.5 (high irrigation) with high intensity in temperate areas and low intensity in tropical areas which reflects the regional economic developments.

Figure 1: Irrigation intensity (Wirrig) changes with mean annual temperature (MAT) as used in the model.

3, section 2.1.5. it says "Crop phenology was not decided beforehand but was determined by the climate condition." Double cropping over the East Asia is very popular where the climate conditions are sufficient. However, in the USA, single cropping is major under similar climate condition. Please discuss this issue a little bit in the discussion section.

Response: Thanks for the suggestion. Indeed multiple cropping is mainly distributed

in low latitude areas in Asia, Africa and South America. We added this part in the discussion part Line 327-331.

4, DO double check the context and the references. There are too many small mistakes in the reference list, such as Subscript 2 in $CO_2$. Mistakes were found in Lines 508 and 518 536: DO keep the references in an identical style.

Response: Thanks for the careful review. We double checked and rectified the context and the references.

[Figure]

**Fig. 1.** Irrigation intensity (Wirrig) changes with mean annual temperature (MAT) as used in the model.

---

## Author Comment (AC2) · 19 Aug 2017

Reviewer 2: This study simulates global cropland NPP from 1961 to 2010 using the VEGAS model and compares the simulation with FAO statistical data on continental and country scales. The comparison indicates general agreement between the model simulation and the statistical data, yet the scientific importance of such comparisons may be questionable. Because the study essentially tuned the model parameters to fit the FAO data, the agreement found in the comparison may only demonstrate the success of the adopted model-tuning methods.

Response: Thanks for the understanding of this paper. The comparisons between the results driven by the default and updated parameters showed the significance of key

parameter calibration at regional scales. Because models may significantly underestimate some regions and overestimate elsewhere even if the global total is simulated correctly, and this cannot present the real carbon pattern (Graven et al., 2013). The regional scale parametrization also constrains the model to provide much more confidence and accuracy in future projections (Le Quéré et al., 2016).

Some of the technical drawbacks of the studies include: 1) The temporal trend of the Green Revolution seems to be totally decided by Eqs. (1) and (5). Are the two equations (which use the same reference year 1960 and the same temporal scale factor of 70 years) representative for all continents/countries under consideration?

Response: Thanks for the question. We acknowledge this limitation due to the lack of detailed data sets for all major regions and countries in long-term scale. For example, the modeled results cannot capture the decreasing trend in the former Soviet Union due to the cropland abandonment after 1990 (Schierhorn et al., 2013), which has not been represented in the data set. Additionally, the Green Revolution was mostly started in the 1960s, and we modeled the first order of such temporal evolution for several decades, thus this simple representation of harvest index and management intensity was good enough to capture most regions. When doing small region/country scale research, the time parameter might be modified based on literature review.

2) How should we interpret/compare the values of M1r? The values in Tables 2 and 3 appear as "magic numbers" to me. For instance, in Table 3 the new M1r values of France and Canada are roughly three times and twice as high as US. What does these values really mean?

Response: Thanks for the careful review. M1r is a region-dependent relative management intensity factor. Generally, M1r is large in highly intensified agricultural areas (e.g., France and China) and small in less management areas (e.g., India and Argentina). M1r is also associated with the growing season length. High latitude areas (e.g., Canada and Northern Europe) with low mean annual temperature (MAT) have

short growing season, in order to take advantage of the short growing season more actively, regional management intensity needs to be higher at high latitudes than low latitudes.

3) Fig. 8 and corresponding text: Avoid the use of "Tg C per 0.5 deg grid cell" as the unit for crop NPP as the area of "0.5 deg grid cell" varies at different latitudes. The results shown in the figure thus are potentially misleading.

Response: Thanks for the suggestion. We revised it accordingly in Lines 304, 307 and 740 using Tg C per 2500 km2.

4) a minor comment: Lines 257-260 on Page 11 state that the adjusted M1r parameters produced "dramatically" different estimations for continents include Oceania (Fig. 5j).Why? It appears to me that the difference in Africa (Fig. 5a) is much more "dramatic" than Oceania.

Response: Thanks for the careful review. We revised the description in the text Line 264. Additionally, when seeing the results from different scales, Oceania indeed showed dramatic improvement (Fig. 1).

Figure 1: The calibrated results captured the Oceania crop productions much better than the default ones.

Otherwise the paper is well written and easy to read.

References:

Graven, H. D., Keeling, R. F., Piper, S. C., Patra, P. K., Stephens, B. B., Wofsy, S. C., Welp, L. R., Sweeney, C., Tans, P. P., and Kelley, J. J.: Enhanced seasonal exchange of CO2 by northern ecosystems since 1960, Science, 341, 1085-1089, 2013. Le Quéré, C., Andrew, R. M., Canadell, J. G., Sitch, S., Korsbakken, J. I., Peters, G. P., Manning, A. C., Boden, T. A., Tans, P. P., Houghton, R. A., Keeling, R. F., Alin, S., Andrews, O. D., Anthoni, P., Barbero, L., Bopp, L., Chevallier, F., Chini, L. P., Ciais, P., Currie, K., Delire, C., Doney, S. C., Friedlingstein, P., Gkritzalis, T., Harris, I., Hauck,

J., Haverd, V., Hoppema, M., Klein Goldewijk, K., Jain, A. K., Kato, E., Körtzinger, A., Landschützer, P., Lefèvre, N., Lenton, A., Lienert, S., Lombardozzi, D., Melton, J. R., Metzl, N., Millero, F., Monteiro, P. M. S., Munro, D. R., Nabel, J. E. M. S., Nakaoka, S. I., O'Brien, K., Olsen, A., Omar, A. M., Ono, T., Pierrot, D., Poulter, B., Rödenbeck, C., Salisbury, J., Schuster, U., Schwinger, J., Séférian, R., Skjelvan, I., Stocker, B. D., Sutton, A. J., Takahashi, T., Tian, H., Tilbrook, B., van der Laan-Luijkx, I. T., van der Werf, G. R., Viovy, N., Walker, A. P., Wiltshire, A. J., and Zaehle, S.: Global Carbon Budget 2016, Earth Syst. Sci. Data, 8, 605-649, 2016. Schierhorn, F., Müller, D., Beringer, T., Prishchepov, A. V., Kuemmerle, T., and Balmann, A.: Post‐Soviet cropland abandonment and carbon sequestration in European Russia, Ukraine, and Belarus, Global Biogeochemical Cycles, 27, 1175-1185, 2013.

[Figure]

[Figure]

**Fig. 1.** The calibrated results captured the Oceania crop productions much better than the default ones.